# Assessment of compliance with quality laboratory standards in medical laboratories within the Kumasi Metropolis, Ghana

Elizabeth Sorvor[1]*, Mawuli Dzodzomenyo[2], Justice Nonvignon[3], Genevieve Cecilia Aryeetey [3]*

1 Kumasi South Hospital, Ghana Health Service, Ghana, 2 Department of Biological, Environmental and Occupational Health, University of Ghana, School of Public Health, Ghana, 3 Department of Health Policy, Planning and Management, University of Ghana, School of Public Health, Ghana

* mylisas2@yahoo.com (ES); gcaryeetey@ug.edu.gh (GCA)

## Abstract

### Background

In Ghana, medical laboratories are a vital component of the health system, but there has been limited evidence on how well they meet international quality standards. This study assessed compliance with laboratory standards by medical laboratories in the Kumasi Metropolis of Ghana.

### Methods

A descriptive cross-sectional survey was conducted among forty-three (43) laboratory facilities, including those operating within healthcare facilities and private standalone laboratories, in the Kumasi metropolis from 8th March to 26th November, 2021. The healthcare facilities assessed include seven (7) government, four (4) Christian Health Association of Ghana (GHAG), twenty-one (21) private hospitals, and eleven (11) standalone laboratories. The World Health Organisation Stepwise Laboratory Quality Improvement Process Towards Accreditation (WHO SLIPTA) checklist was used to assess compliance with laboratory standards. The checklist comprises 12 main sections, containing 117 questions, resulting in a total of 275 scores. Each item is awarded a point value of 2, 3, or 5 points based on relative importance and complexity. Scores and percentages of compliance are categorized as follows: 0–150 (<55%), 151–177 (55–64%), 178–205 (65–74%), 206–232 (75–84%), 233–260 (85–94%), and 261–275 (≥95%). These categories ultimately translate to a rating of 0–5 stars. Descriptive analysis and the Kruskal-Wallis test were conducted to examine the differences in overall compliance scores across the types of health facilities.

**Data availability statement:** All relevant data are within the manuscript and its Supporting Information files.

**Funding:** The author(s) received no specific funding for this work.

**Competing interests:** The authors have declared that no competing interests exist.

## Results

The overall median compliance score was 61(interquartile range 53–66), the minimum score was 49, and the maximum score was 104. CHAG facilities had significantly higher scores than private facilities (Z = 2.53, p = 0.03) and standalone facilities (Z = 3.60, p < 0.001). Similarly, scores for government facilities were significantly higher than standalone facilities (Z = 3.30, p < 0.001).

## Conclusion

All facilities had a zero (0) star rating and failed to meet the minimum compliance level of 55%.

## Introduction

Medical laboratory services have evolved to become an integral part of healthcare delivery systems worldwide [1, 2]. The role of the medical laboratory is recognized as critical in the general improvement of countries' health systems [3]. The assertion that " seventy percent (70%) of clinical decisions depend on laboratory data" has become increasingly undeniable [4]. However, evidence suggests that 47% of the world's population lacks access to diagnostic services, with rates ranging from 35% to 62% in low- and middle-income countries (LMICs) [5].

As healthcare systems advance and global health risks grow, quality medical laboratory services are vital for ensuring high-quality healthcare provision [6, 7]. Lack of access to quality diagnostics is one of the most significant causes of unsafe health [8]. Quality assurance in medical laboratories is a global necessity, significantly impacting patient outcomes and disease surveillance [5, 9]. As healthcare systems face growing patient demands, the resurgence of diseases, the introduction of new technologies, and evolving regulatory standards, the quality of medical laboratories has become a critical concern [10, 11]. As the demand for diagnostic services continues to surge, laboratories face the imperative to balance increased throughput with uncompromising accuracy and reliability [12, 13]. This necessitates a holistic, systematic approach that encompasses every facet of laboratory operations, including compliance with quality standards [14].

Access to quality medical laboratory services in Africa is a major challenge, leading to delays and inadequate responses to epidemics and to inadequate patient management [8, 15]. Furthermore, the poor quality of laboratory services in Africa exacerbates morbidity and mortality in the region [15]. Concerns about the quality of laboratory testing have led to the establishment of regulations, standard guidelines, and quality improvement programs within the medical laboratory system to reduce testing errors [16].

In 2013, Ghana implemented the Strengthening Laboratory Management Toward Accreditation (SLMTA) initiative in fifteen(15) medical laboratories across the country to enhance the quality of medical laboratory services [17]. This structured training and mentorship framework enables laboratories to meet international quality

standards, thereby improving the accuracy and reliability of diagnostic services, which are essential for patient care and disease surveillance [18]. However, over the years, the progress made by these laboratories through the SLMTA initiative could not be maintained due to a lack of management commitment, which is vital for effectively implementing a laboratory quality management system [17]. Only five (5) out of the eight hundred (800) public sector laboratories, which conduct the majority of patient testing, have recently received accreditation to international standards (Southern African Development Community Accreditation Service [SADCASS] and Quality and Accreditation Institute of India [19, 20]).

Ghana's health policy and quality strategy report indicates that healthcare quality at all levels and across all services is poor, with quality strategies having little impact on patient experiences and health outcomes [19, 20]. The healthcare system continues to be plagued by varying degrees of inefficiencies in medical laboratory diagnostics, and in some instances, laboratories are not equipped to respond to local health needs. As Ghana's healthcare system continues to evolve, ensuring the sustainability of medical laboratory quality is a critical concern that requires rigorous assessment to provide evidence needed to address gaps in service quality. However, there has been limited evidence on how well medical laboratories comply with international quality standards. Previous studies have primarily focused on a small number of laboratories [21, 22]. As a result, existing research is inadequate to address problems associated with the medical laboratory setup in Ghana. Medical laboratories, therefore, find themselves ill-equipped to address issues related to quality laboratory services.

Access to high-quality laboratory services is essential for achieving Universal Health Coverage (UHC) and the Sustainable Development Goals (SDGs) (Goal 3) [15, 23]. Evaluating compliance with quality laboratory standards in medical laboratories is crucial, particularly the comprehensive analysis of quality system essentials and human resource capacity, which are building blocks of laboratory quality. The study aims to evaluate compliance with quality laboratory standards in one of Ghana's largest metropolitan areas in the Ashanti region. This approach will characterize and quantify laboratory service quality gaps to inform the formulation of policies and interventions to strengthen Ghana's medical laboratory systems for quality healthcare.

## Methods

### Study design and setting

The study employed a descriptive cross-sectional design. The study was conducted in the Kumasi Metropolis, Ashanti Region, Ghana. The region's unique central position makes it accessible from all corners of the country. Kumasi is the second-largest city in the country and the administrative capital of the Ashanti region [24]. According to the District Health Information Management System II (DHIMSII), the metropolis has about two hundred and eighty-three (283) healthcare facilities, made up of hospitals, health centres, and clinics. One hundred and thirty-nine (139) government, one hundred and thirty-three (133) private and eleven (11) CHAG. Additionally, according to the Association of Private Medical Laboratories (APML), there are twenty-five (25) standalone private medical laboratories in the metropolis. However, only hospitals that provide medical laboratory services and standalone medical laboratories were recruited for the study. Kumasi was chosen for the study because it serves as a major referral centre for adjoining communities and provides a blend of all healthcare facilities. Thus, it is expected to provide high-quality healthcare for the population.

### Study population and sample

The study population was hospitals and standalone medical laboratories. There was a total of eighty-one (81) healthcare facilities: forty-seven (47) private, seven (7) government, four (4) CHAG hospitals, and twenty-five (25) standalone laboratories. A Census of all medical laboratories in hospitals and standalone medical laboratories in the Kumasi metropolis was conducted to assess compliance with quality laboratory standards.

## Instrument

The WHO SLIPTA checklist was adapted to measure compliance with laboratory standards. The elements of this checklist are based on ISO 15189:2012 and Clinical Laboratory Standards Institute (CLSI) guidelines on Quality Management System: A Model for Laboratory Services [25]. It is a requirement for laboratories to demonstrate the quality and reliability of their services. The checklist assessment enables the determination of whether a laboratory is providing accurate and reliable results, is well-managed and adhering to good laboratory practices, and identifies areas for improvement [25].

The checklist comprises 12 main sections, with 117 questions and a total of 275 scores. Each item is awarded 2, 3, or 5 points, based on relative importance and complexity. Scores and percentages of compliance are categorised as follows: 0–150 (<55%), 151–177 (55–64%), 178–205 (65–74%), 206–232 (75–84%), 233–260 (85–94%), and 261–275 (≥95%). These categories ultimately translate to a rating of 0–5 stars (Table 1).

## Data collection

The laboratory assessment was conducted from 8th March to 26th November, 2021. Out of the 81 expected health facilities, only forty-three (43) facilities participated in the study; twenty-six (26) private hospitals and fourteen (14) standalone medical laboratories declined to participate. Laboratory assessment was conducted by an African Society of Laboratory

**Table 1. Items Included in the Questionnaire and Score Sheets.**

| Assessment Score Sheet | |
| --- | --- |
| **Section** | **Total points** |
| **Section 1:** Documents & Records | 28 |
| **Section 2:** Management Reviews | 14 |
| **Section 3:** Organization & Personnel | 22 |
| **Section 4:** Client Management & Customer Service | 10 |
| **Section 5:** Equipment | 35 |
| **Section 6:** Evaluation and Audits | 15 |
| **Section 7:** Purchasing & Inventory | 24 |
| **Section 8:** Process Control | 32 |
| **Section 9:** Information Management | 21 |
| **Section 10:** Identification of Non-Conformities, Corrective and Preventive Actions(NCPA) | 19 |
| **Section 11:** Occurrence/Incident Management & Process Improvement | 12 |
| **Section 12:** Facilities and Biosafety | 43 |
| **TOTAL SCORE** | 275 |

| No Stars (0–150 pts) <55% | 1 Star (151–177pts) 55-64% | 2 Star (178–205 pts) 65-74% | 3 Star (206–232 pts) 75-84% | 4 Star (233–260 pts) 85-94% | 5 Star (261-275) ≥95% |
| --- | --- | --- | --- | --- | --- |

Medicine (ASLM) certified laboratory auditor. The assessment was announced during the facility entry process, before data collection commenced. Data were collected in accordance with the checklist requirements, using a structured questionnaire, observation, and document review. The checklist was administered to each laboratory by reviewing laboratory documents to verify that the laboratory quality manual, policies, Standard Operating Procedures (SOPs), and other manuals were complete, current, accurate, and reviewed annually. Additionally, laboratory records, including equipment maintenance records, audit trails, incident reports, logs, personnel files, Internal Quality Control (IQC) records, and External Quality Assurance records (EQA), were reviewed. Additionally, laboratory operations were closely monitored to ensure compliance with established written policies and procedures throughout the pre-analytic, analytic, and post-analytic phases of testing. It was confirmed that the laboratory procedures are suitable for the tests being conducted and that any identified deficiencies and nonconformities are thoroughly investigated and addressed within the designated timeframe [11]. The health sector staffing norm, a workload-related human resource planning and management tool that provides the number and calibre of healthcare personnel required in a given health facility [26], was used to determine the staffing requirements for each laboratory staff category within a health facility type. The outpatients per year for each facility type were used to determine the required number of medical laboratory professionals in each facility, as stipulated in the tool. In private hospitals and standalone laboratories, the low volume of outpatient attendance affected the categorization of Medical Laboratory Scientists, as this is not required within their workload category.

### Data analysis

Data were imputed in Microsoft Excel sheets, coded, and analysed using Stata version 14 (StataCorp LLC). Overall compliance was calculated as the sum of the scores for the 12 Quality System Essentials (QSEs) assessed in each laboratory.

The data's normality was assessed using the Kolmogorov–Smirnov test, which rejected the normality hypothesis ($P < 0.001$). Consequently, nonparametric tests were performed on the data. Descriptive statistics, including raw scores, median, average, and percentages of the maximum score and star equivalents, were used to analyse compliance with the quality laboratory standard. Additionally, the Kruskal-Wallis Test was conducted to examine differences in median compliance scores across types of health facilities. Dunn's pairwise comparison test was conducted following a significant Kruskal–Wallis tests to determine the magnitude of the differences in compliance among types of health care facilities.

### Ethical consideration

This study is part of the PhD research work of the first author, and received ethical approval from the Ghana Health Service Ethics Review Committee (Reference #: GHS-ERC: 014/11/20) and Komfo Anokye Teaching Hospital Institutional Review Board (IRB) (Reference #: KATH IRB/AP/155/20). Permission was obtained from the Kumasi Metro Health Directorate and the Municipal Health Directorate of the selected health facilities. The privacy and confidentiality of study participants were ensured by safeguarding the collected data and using codes instead of facility names.

## Results

### Characteristics of health facilities and human resource profile

A total of forty-three (43) medical laboratories were assessed. The majority (72%) of laboratories were privately owned health facilities (private hospitals and laboratories), and the least (4%) were government-owned tertiary and secondary hospitals (Table 2). There was a deficit of 51 medical laboratory scientists in government-owned tertiary health facilities and 8 in secondary health facilities. However, there was a surplus of medical laboratory scientists in primary (4), private hospitals (23) and CHAG (7) facilities (Table 3).

**Table 2. Background Characteristics of Health Facilities.**

| Facility ownership | Facility type | N (%) |
|---|---|---|
| Government | Tertiary | 1(2.3) |
| | Secondary | 1 (2.3) |
| | Primary | 5(11.7) |
| CHAG | Primary | 4(9.3) |
| Private | Primary | 21(48.8) |
| | Laboratory | 11(25.6) |
| **Total** | | **43(100)** |

**Table 3. Human resource profile of health facilities.**

| Type of personnel | Facility type | Expected number | Number available | Difference | Interpretation |
|---|---|---|---|---|---|
| Medical Laboratory Scientist | Tertiary | 127 | 76 | −51 | Deficit |
| | Secondary | 12 | 4 | −8 | Deficit |
| | Primary | 28 | 32 | +4 | Surplus |
| | *CHAG | 13 | 20 | +7 | Surplus |
| | *Private hospital | 11 | 34 | +23 | Surplus |
| | Private Laboratory | 0 | 16 | +16 | Surplus |
| | | | | | |
| Medical Laboratory Technician | Tertiary | 0 | 28 | +28 | Surplus |
| | Secondary | 8 | 5 | −3 | Deficit |
| | Primary | 41 | 17 | −24 | Deficit |
| | *CHAG | 29 | 17 | −12 | Deficit |
| | *Private Hospital | 30 | 29 | −1 | Deficit |
| | *Private Laboratory | 1 | 16 | +15 | Surplus |
| | | | | | |
| Medical Laboratory Assistant | Tertiary | 0 | 1 | +1 | Surplus |
| | Secondary | 0 | 2 | +2 | Surplus |
| | Primary | 27 | 8 | −19 | Deficit |
| | *CHAG | 24 | 12 | −12 | Deficit |
| | *Private hospital | 42 | 12 | −30 | Deficit |
| | *Private Laboratory | 12 | 3 | −9 | Deficit |
| | | | | | |
| Phlebotomist | Tertiary | 23 | 10 | −13 | Deficit |
| | Secondary | 15 | 0 | −15 | Deficit |
| | Primary | 0 | 1 | +1 | Surplus |
| | *CHAG | 0 | 0 | 0 | Adequate |
| | *Private hospital | 0 | 0 | 0 | Adequate |
| | *Private Laboratory | 0 | 2 | + 2 | Surplus |

**Key *=Estimated expected number of personnel: public health sector staffing norms were adapted for CHAG and private facilities, which affected the interpretation of deficits and surpluses, particularly in private hospitals and standalone laboratories.**

## Laboratory compliance scores by facility type

Out of the expected score of 275, the overall median compliance score across facilities was 61 (interquartile range 53–66); the minimum score was 49, and the maximum score was 104. Tertiary-level facility laboratories performed better

(93) compared with secondary (66) and Primary (64.8) level facilities. Similarly, faith-based organisations (CHAG) performed better (81.1) than Private hospitals (60.1) and Private Standalone laboratories (54.3) (Tables 4 and 5). All laboratories performed poorly on the twelve quality system essentials (QSE) across health facilities (Table 6). However, facilities and biosafety showed better performance across facility types, particularly in government (16.9) and CHAG (16.8) health facilities. Similarly, equipment management scored higher in CHAG (14.3) than in government (10.4), private hospitals (9.0), and standalone laboratories (7.4). The lowest-performing QSEs were: document and records; management review and responsibility; Client Management and Customer Service; Evaluation and Audits; Identification of Nonconformities; Corrective Action and Preventive Actions; and Occurrence Management.

**Differences in laboratory compliance score by facility type**

A Kruskal-Wallis Test was conducted to examine differences in overall compliance scores and quality system essential scores across types of health facilities (Table 7). There were significant differences in the overall median compliance score (p < 0.001) among the four [4], categories of health facilities. However, Dunn's pairwise post hoc comparison with Bonferroni adjustment test (Table 8) showed that CHAG facilities had significantly higher scores than private facilities (Z = 2.53, p = 0.03) and standalone facilities (Z = 3.60, p < 0.001). Similarly, scores for government facilities were significantly higher than standalone facilities (Z = 3.30, p < 0.001).

Similarly, there were significant differences (Table 7) in the following QSEs across facility types; organization and personnel (p < 0.001), equipment management (p = 0.03), process control (p = 0.04) and information management (p = 0.01). Furthermore, in the pairwise comparisons analysis (Table 8) Organization & Personnel scores for government facilities were significantly higher than private facilities (Z = 3.17, p < 0.001) and standalone facilities (Z = 3.88, p < 0.001). Also, scores for CHAG facilities were significantly higher than those of private facilities (Z = 2.45, p = 0.04) and standalone facilities (Z = 3.13, p = 0.01). Equipment management scores for CHAG facilities were significantly higher compared to standalone facilities (Z = 3.00, p = 0.01); process control scores for government facilities were significantly higher than those of standalone facilities (Z = 2.90, p = 0.01) and CHAG facilities also had significantly higher scores in information management compared to standalone facilities (Z = 3.24, p < 0.001).

## Discussion

The study revealed low compliance with quality laboratory standards across various health facilities. However, government and CHAG facilities recorded significantly higher overall compliance scores than private and standalone facilities, demonstrating quality system gaps at lower levels of care. These disparities underscore the need for targeted quality-improvement interventions, particularly in private hospitals and standalone laboratories. No facility achieved the minimum expected compliance score, highlighting systemic weaknesses in laboratory quality management systems in the healthcare sector. This low compliance has been identified in previous studies conducted in similar settings [21, 26, 27].

Furthermore, the study found that the performance of all laboratories in the 12 QSEs was below the required standard. However, across all types of health facilities, compliance was relatively stronger for biosafety and equipment management QSEs. Similar trends have been reported in other studies [21] [27] where laboratories prioritise tangible quality indicators over system-based processes. In contrast, Management Review and Responsibility, Evaluation and Audits, Occurrence Management, and Identification of Nonconformities and Corrective and Preventive Actions (NCCPA) were almost absent across all settings. These findings corroborate studies from similar settings [28] [21]. These QSEs are central to ISO 15189 requirements for continual improvement, and their absence poses a significant challenge to sustainable quality laboratory service and patient safety. Hence, targeted interventions are required to build capacity in quality management systems, particularly in primary-level facilities and private standalone laboratories.

**Table 4. Overall compliance score by each laboratory.**

| Facility | Ownership | Compliance score |
| --- | --- | --- |
| CHG01 | CHAG | 84 |
| CHG02 | CHAG | 66 |
| CHG03 | CHAG | 104 |
| CHG04 | CHAG | 69 |
| G001 | Government | 63 |
| G002 | Government | 67 |
| G003 | Government | 66 |
| G004 | Government | 64 |
| G005 | Government | 62 |
| G006 | Government | 68 |
| G007 | Government | 93 |
| P001 | Private | 72 |
| P002 | Private | 62 |
| P003 | Private | 62 |
| P004 | Private | 80 |
| P005 | Private | 74 |
| P006 | Private | 58 |
| P007 | Private | 59 |
| P008 | Private | 68 |
| P009 | Private | 53 |
| P010 | Private | 59 |
| P011 | Private | 57 |
| P012 | Private | 53 |
| P013 | Private | 61 |
| P014 | Private | 55 |
| P015 | Private | 55 |
| P016 | Private | 56 |
| P017 | Private | 53 |
| P018 | Private | 54 |
| P019 | Private | 50 |
| P021 | Private | 64 |
| P022 | Private | 57 |
| S001 | Standalone | 63 |
| S002 | Standalone | 54 |
| S003 | Standalone | 50 |
| S004 | Standalone | 50 |
| S005 | Standalone | 49 |
| S006 | Standalone | 49 |
| S007 | Standalone | 64 |
| S008 | Standalone | 49 |
| S009 | Standalone | 52 |
| S010 | Standalone | 66 |
| S011 | Standalone | 51 |

**Table 5. Laboratory compliance score by facility type.**

| Facility characteristics | Laboratory compliance score (%) | Overall expected score | Change | Star Rating |
|---|---|---|---|---|
| Tertiary | 93.0(33.8) | 275 | 182.0 | 0 |
| Secondary | 66.0(24) | 275 | 209.0 | 0 |
| Primary | 64.8(23.6) | 275 | 210.2 | 0 |
| CHAG | 81.1(29.5) | 275 | 193.9 | 0 |
| Private hospital | 60.1(21.9) | 275 | 214.9 | 0 |
| Private laboratory | 54.2(19.7) | 275 | 220.8 | 0 |

Notation: Raw scores for Tertiary and Secondary facilities; average score for primary, CHAG and private health facilities.

**Table 6. Laboratory Performance in the Quality System Essentials by Facility Type.**

| Quality system essentials | Government | | | Govt. | CHAG | Private | | Expected scores |
|---|---|---|---|---|---|---|---|---|
| | Tertiary | Secondary | Primary | | | Hospital | Laboratory | |
| Document & Records | 4.0 | 2.0 | 2.0 | 3.0 | 2.8 | 2.2 | 2.0 | 28 |
| Management Review & Responsibility | 0 | 0 | 0 | 0.0 | 0.3 | 0 | 0 | 14 |
| Organization & Personnel | 11.0 | 10.0 | 8.2 | 9.7 | 8.5 | 5.8 | 4.9 | 22 |
| Client Management and Customer Service | 2.0 | 3.0 | 2.0 | 2.3 | 3.5 | 2.7 | 2.3 | 10 |
| Equipment | 15.0 | 7.0 | 9.2 | 10.4 | 14.3 | 9.0 | 7.4 | 35 |
| Evaluation & Audits | 0 | 0 | 0 | 0.0 | 0 | 0 | 0 | 15 |
| Purchasing &Inventory | 11.0 | 10.0 | 13.0 | 11.3 | 13.8 | 11.9 | 10.5 | 24 |
| Process control | 15.0 | 9.0 | 9.0 | 11.0 | 8.5 | 7.5 | 6.5 | 32 |
| Information management | 10.0 | 8.0 | 7.6 | 8.5 | 11.0 | 7.7 | 6.8 | 21 |
| Identification of NCPA* | 1.0 | 0 | 0 | 0.3 | 0.3 | 0 | 0 | 19 |
| Occurrence management | 4.0 | 0 | 0 | 1.3 | 1.3 | 0.0 | 0 | 12 |
| Facilities & Biosafety | 20.0 | 17.0 | 13.8 | 16.9 | 16.8 | 13.2 | 13.8 | 43 |
| **Overall compliance score** | **93.0** | **66.0** | **64.8** | **74.7** | **81.1** | **60.1** | **54.3** | **275** |

Notation: Raw scores for Tertiary and Secondary facilities; average score for Primary, CHAG and private health facilities.

The WHO indicate that although implementing quality management does not guarantee an error-free laboratory, it helps ensure high-quality laboratory services that detect and prevent errors from recurring [29]. It is, therefore, imperative that medical laboratories implement a quality management system to minimise errors and maintain high-quality laboratory services. Inadequate quality has been identified as a barrier to the provision of optimal laboratory services in low- and middle-income countries [23]. The current study shows that compliance with quality laboratory standards falls short of the required standards; leadership and management accountability must be strengthened to ensure laboratory quality at all levels of care. Medical laboratories operate in a complex environment, and ensuring high-quality services requires rigorous implementation of standards and guidelines [2].

This study reveals a significant gap between actual laboratory staffing levels and the standards set by Ghana's Health sector. It highlights challenges in equitably distributing human resources for laboratory services. While some facility categories have sufficient personnel, the distribution across different levels and ownership types falls short of expected norms. This imbalance affects equitable access to quality laboratory services and the efficiency of diagnostics in the healthcare system. Previous studies have corroborated this finding, particularly inadequate numbers and skewed distribution [30] [23]. These findings suggest that the current staffing patterns are influenced more by institutional and market dynamics than by evidence-based workforce planning. Enhancing the use of Ghana's health sector staffing norms, through tools like

**Table 7. Differences in Laboratory Compliance Score by Facility Type.**

| Assessment type | Facility type | Quality score | |
|---|---|---|---|
| | | **Median (IQR)** | **p-value** |
| Document & Records | Government | 2 (2 2) | 0.54 |
| | CHAG | 2.5(2,3.5) | |
| | Private hospital | 2 (2 2) | |
| | Private laboratory | 2 (2 2) | |
| | | | |
| Management Review & Responsibility | Government | 0(0,0) | 0.88 |
| | CHAG | 0(0.5) | |
| | Private hospital | 0(0,0) | |
| | Private laboratory | 0(0,0) | |
| | | | |
| Organization & Personnel | Government | 8 (8 10) | 0.00* |
| | CHAG | 8 (8 9) | |
| | Private hospital | 6(5.7) | |
| | Private laboratory | 5 (4 6) | |
| | | | |
| Client Management and Customer Service | Government | 2 (2 3) | 0.06 |
| | CHAG | 3.5 (3 4) | |
| | Private hospital | 3 (2 3) | |
| | Private laboratory | 2 (2 3) | |
| | | | |
| Equipment | Government | 8 (7 15) | 0.03 |
| | CHAG | 12.5(10.5,18) | |
| | Private hospital | 8 (7 10) | |
| | Private laboratory | 8 (6 10) | |
| | | | |
| Evaluation & Audits | Government | 0(0,0) | 1.00 |
| | CHAG | 0(0,0) | |
| | Private hospital | 0(0,0) | |
| | Private laboratory | 0(0,0) | |
| | | | |
| Purchasing & Inventory | Government | 12 (11 14) | 0.07 |
| | CHAG | 14(12,15.5) | |
| | Private hospital | 11 (10 13) | |
| | Private laboratory | 11 (9 12) | |
| | | | |
| Process control | Government | 9 (8 12) | 0.04 |
| | CHAG | 8.5 (5 12) | |
| | Private hospital | 8 (6 8) | |
| | Private laboratory | 7 (5 7) | |
| | | | |
| Information management | Government | 8 (7 10) | 0.01 |
| | CHAG | 11.5 (9 13) | |
| | Private hospital | 7 (7 8) | |
| | Private laboratory | 6 (6 7) | |
| | | | |

*(Continued)*

**Table 7.** (Continued)

| Assessment type | Facility type | Quality score | |
|---|---|---|---|
| | | **Median (IQR)** | **p-value** |
| Identification of Non-Conformities Corrective & Preventive Action | Government | 0(0,0) | 0.83 |
| | CHAG | 0(0,0.5) | |
| | Private hospital | 0(0,0) | |
| | Private laboratory | 0(0,0) | |
| | | | |
| Occurrence management | Government | 0(0,0) | 0.49 |
| | CHAG | 1(0,2.5) | |
| | Private hospital | 0(0,0) | |
| | Private laboratory | 0(0,0) | |
| | | | |
| Facilities & Biosafety | Government | 15 (12 17) | 0.15 |
| | CHAG | 15.5(14.5,19) | |
| | Private hospital | 13 (11 15) | |
| | Private laboratory | 13 (11 15) | |
| **Overall compliance** | | | |
| | Government | 66(63,68) | 0.00* |
| | CHAG | 76.5(67.5,94) | |
| | Private hospital | 58(55,62) | |
| | Private laboratory | 51(49,63) | |

**Significant level: p<0.001**

**Table 8. Non-parametric pairwise multiple comparison of compliance by facility type.**

| Assessment type | Comparison Pair | Z-test | P-value |
|---|---|---|---|
| | | | |
| Overall compliance score | CHAG -Private | 2.53 | 0.03 |
| | CHAG-Standalone | 3.60 | 0.00* |
| | Government – Standalone | 3.30 | 0.00* |
| Organization & Personnel | Government – Private | 3.17 | 0.00* |
| | Government – Standalone | 3.88 | 0.00* |
| | CHAG – Private | 2.45 | 0.04 |
| | CHAG – Standalone | 3.13 | 0.01 |
| Equipment | CHAG – Standalone | 3.00 | 0.01 |
| Process Control | Government – Standalone | 2.9 | 0.01 |
| Information management | CHAG – Standalone | 3.24 | 0.00* |
| | | | |

**Significant level: p<0.001**

the Workload Indicators of Staffing Need (WISN), could facilitate a more rational deployment and redistribution of laboratory personnel. Aligning staffing decisions with normative standards is crucial for improving diagnostic quality, ensuring the efficient use of scarce human resources, and promoting equity within Ghana's health system.

The predominant deficit of phlebotomists observed in the study underscores a significant gap in human resources. Specimen collection is a fundamental function in the laboratory, and inadequate staffing in this critical area can result in workflow bottlenecks, compromised sample quality, and delayed diagnostic outcomes, which, in turn, can indirectly impact clinical decision-making and patient safety. The term "phlebotomy" refers to the procedure of drawing venous blood, and Phlebotomists are healthcare professionals who have received training to collect blood samples [31]. According to Ghana's health sector staffing norm, only regional and tertiary hospitals must have a phlebotomist [32]. The lack of a framework for phlebotomy and the resulting shortage of phlebotomists across the tiers of laboratory practice in Ghana's health sector are grave deficiencies. The need for phlebotomists in the diagnostic process is highlighted because 60% of laboratory errors occur due to improper methods for collecting venous blood samples during laboratory testing [33]. According to Waheed et al. (2013), even the most advanced laboratory equipment cannot yield accurate results from a specimen collected improperly [34]. Hence, phlebotomists should be considered an essential component of the healthcare system, as they are important in providing suitable laboratory specimens for testing [31]. Mbah (2014) observed that employee cross-training for numerous duties is required in most healthcare facilities across Africa due to resource constraints, potentially eroding specialised skills and leading to excessive workloads [35]. However, in his assessment of phlebotomy and quality in the African laboratory, he argued that using skilled phlebotomy-specific laboratory personnel could significantly reduce pre-analytical error rates in the laboratory diagnostic value chain [35].

Therefore, there is an urgent need to implement pragmatic measures to address the quality gap and related factors that contribute to the provision of quality medical laboratory services in Africa, particularly in Ghana.

## Strength and limitation

The SLIPTA model provided a standardized, structured framework for evaluating compliance with quality laboratory standards. Aligning this model with Ghana's health-sector staffing norms enabled a comprehensive analysis of human resource capacity in medical laboratories.

The study's limitation is that the adaptation of the norms for CHAG and private facilities affected the interpretation of staffing deficits and surpluses, particularly in private hospitals and standalone laboratories. Additionally, the multiple comparisons of quality system indicators across facility types may inflate the type 1 error. The use of the Kumasi Metropolis alone, an urban area, might affect the representativeness and generalizability of the study's findings.

## Conclusion

None of the 43 medical laboratories assessed met or exceeded the minimum quality standards. The study also revealed an uneven distribution of laboratory personnel across different facility types, and most laboratories lacked phlebotomists. Medical laboratories have lagged in adopting modern quality management techniques, which are essential for enhancing the quality of laboratory services. The poor compliance of medical laboratories with basic quality standards could significantly affect the accuracy, reliability and timeliness of laboratory services.

## Supporting information

**S1 File. Laboratory Assessment Data_PLOSONE.**
(XLSX)

## Acknowledgments

We would like to thank the research team, all hospitals, Health Directors, Health Service Administrators, Medical Superintendents, Nurse Managers, and Laboratory Managers who assisted with the recruitment of study participants and data collection.

## Author contributions

**Conceptualization:** Elizabeth Sorvor.

**Data curation:** Elizabeth Sorvor.

**Formal analysis:** Genevieve Cecilia Aryeetey, Elizabeth Sorvor, Mawuli Dzodzomenyo, Justice Nonvignon.

**Investigation:** Elizabeth Sorvor.

**Methodology:** Genevieve Cecilia Aryeetey, Elizabeth Sorvor, Mawuli Dzodzomenyo, Justice Nonvignon.

**Project administration:** Elizabeth Sorvor.

**Resources:** Elizabeth Sorvor.

**Software:** Elizabeth Sorvor.

**Supervision:** Genevieve Cecilia Aryeetey, Mawuli Dzodzomenyo, Justice Nonvignon.

**Validation:** Genevieve Cecilia Aryeetey, Mawuli Dzodzomenyo, Justice Nonvignon.

**Writing – original draft:** Elizabeth Sorvor.

**Writing – review & editing:** Genevieve Cecilia Aryeetey, Elizabeth Sorvor, Mawuli Dzodzomenyo, Justice Nonvignon.

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
