## [Decision Letter · Decision Letter 0]

8 Jan 2026

PONE-D-25-53279Assessment of compliance with quality laboratory standards in medical laboratories within the Kumasi Metropolis, GhanaPLOS One

Dear Dr. Aryeetey,

Thank you for submitting your manuscript to PLOS ONE. After careful consideration, we feel that it has merit but does not fully meet PLOS ONE’s publication criteria as it currently stands. Therefore, we invite you to submit a revised version of the manuscript that addresses the points raised during the review process.

Indicate the corresponding author for the publicationIn the abstract, indicate how many of the facilities were public and how many were private standalone. In addition, indicate all the classes of the health facilities, tertiary, secondary or primaryDefine all abbreviations in the abstract on first mentionIn the method section, define how the scoring was doneState the p-value as p<0.001.The findings did not support the conclusion since you did not state what a desirable score was

Introduction

Define all abbreviations on first mentionI struggled to find ref number 10 online, and for that matter, the claim that Reports suggest that Ghana's healthcare services are of poor quality, impacting patient experiences and health outcomes, cannot be substantiated. In addition, you mention reports but cited only one, which could not even be verified.Line 63: mention the accreditation institutionsThe gap necessitating this study was missing. Authors should explicitly state the problem being solved by this study

Results

Table 3: Does it mean that no MLS was expected in a private laboratoryLine 158: Tertiary-level facility laboratories performed better (93) compared with secondary (66) and Primary (64.8) level facilities. Provide a statistical decision for this statement.Table 4, could you perform ANOVA and a multiple comparison test to determine the best performing class of facilityTable 5: Some of the columns are not self-explanatory. Indicate if some of the values are average scoresWrite the p-values as <0.001

Discussion

Revise the discussion per the adjustments made in the results

Other

Check the spelling in reference number 10

We look forward to receiving your revised manuscript.

Kind regards,

Enoch Aninagyei, PhD

Academic Editor

PLOS One

Journal Requirements:

2. Please include captions for your Supporting Information files at the end of your manuscript, and update any in-text citations to match accordingly. Please see our Supporting Information guidelines for more information: http://journals.plos.org/plosone/s/supporting-information .

Reviewers' comments:

Reviewer's Responses to Questions

**Comments to the Author**

1. Is the manuscript technically sound, and do the data support the conclusions?

Reviewer #1: Yes

Reviewer #2: Yes

Reviewer #3: Yes

Reviewer #4: Yes

Reviewer #5: Yes

2. Has the statistical analysis been performed appropriately and rigorously? 

Reviewer #1: No

Reviewer #2: Yes

Reviewer #3: Yes

Reviewer #4: Yes

Reviewer #5: Yes

3. Have the authors made all data underlying the findings in their manuscript fully available?

Reviewer #1: Yes

Reviewer #2: Yes

Reviewer #3: Yes

Reviewer #4: Yes

Reviewer #5: Yes

4. Is the manuscript presented in an intelligible fashion and written in standard English?

Reviewer #1: No

Reviewer #2: Yes

Reviewer #3: Yes

Reviewer #4: Yes

Reviewer #5: Yes

5. Review Comments to the Author

Reviewer #1: 1. It is unclear how the authors arrived at the inclusion of forty-three (43) laboratory facilities in the study. This information is mentioned only in the abstract, with no corresponding explanation in the Methods section.

2. References should be added to support the information provided in the Methods section, specifically under Study Design and Setting (page 3).

3. In-text references should be formatted correctly throughout the manuscript. For example, a space is required between the word and the citation number (e.g., [17]).

4. Typographical and formatting errors should be corrected throughout the manuscript. Examples include:

o “Organisation” in Table 1 (page 5)

o Statistical notation in the Data Analysis section (e.g., P < .001, page 6 as 0.001)

5. In Table 3, the calculated differences for surplus and deficit should include the corresponding percentages in parentheses. While percentages are discussed in the text (pages 5–6), their absence from the table makes it inconvenient for readers to recalculate the values.

6. The manuscript reports a minimum compliance score of 49 (page 8, line 157); however, Table 4 lists a score of 48 for facility P019. The supplemental spreadsheet indicates a score of 50 for this facility. This discrepancy suggests a typographical or data export error. Please correct and ensure consistency across all tables and supplementary materials.

7. In Table 4, please indicate whether each facility is primary, secondary, or tertiary under the Ownership column (e.g., in parentheses). Otherwise, this information is only described in the text (page 8, lines 158–161).

8. Overall, the Discussion section is poorly structured and lacks sufficient depth. The conventional approach is to first present and interpret the study’s findings, then compare them with previously published literature not the reverse. The discussion should focus primarily on the manuscript’s main findings rather than on prior studies.

Additionally, several ideas that belong in the same paragraph are presented separately despite conveying similar messages. For example, the discussion of Agboli et al. (2018) and the subsequent paragraph beginning with “A recent survey conducted by Mulleta et al.” both address the failure to meet minimum compliance scores. These should be integrated and presented as one paragraph.

9. Table 1 appears to present less critical information and does not substantially contribute to the main narrative. It would be more appropriate to move this table to the supplementary materials.

10. In the supplemental/supporting spreadsheet, please clearly indicate whether each facility is primary, secondary, or tertiary to facilitate cross-checking with Table 2.

11. The data presented on page 8 (lines 163–168) are difficult to verify using Table 5. It is unclear how the reported values were derived. For example, the score of 3 for Client Management and Customer Service (3) cannot be readily traced to Table 5. Please rewrite this section to clearly align with and directly reference the data presented in the table.

Reviewer #2: Review Comments

Manuscript ID: PONE-D-25-53279

Title: Assessment of compliance with quality laboratory standards in medical laboratories within the Kumasi Metropolis, Ghana

Journal: PLOS ONE

General Assessment

The manuscript evaluates compliance with quality laboratory standards among medical laboratories in the Kumasi Metropolis using the WHO SLIPTA checklist. This is a highly relevant topic for laboratory system strengthening in low- and middle-income countries and aligns well with the scope of PLOS ONE.

Key strengths include the use of a recognized international assessment framework and the attempt to assess all eligible laboratories within the study area. Nonetheless, important aspects of the Methods, Results, and Discussion sections require clearer explanation to ensure that readers can fully understand how scores were generated, how facilities were selected, and how conclusions were derived from the data.

Major Comments

1. SLIPTA Scoring, Categories, and Interpretation

The manuscript reports overall and domain-specific compliance scores (e.g., a median score of 61) without clearly explaining how these relate to the maximum SLIPTA score of 275 or to the official SLIPTA star-rating system. For readers unfamiliar with SLIPTA, the practical meaning of these values is difficult to interpret.

In addition, the manuscript introduces custom compliance categories (low, moderate, high) that do not directly correspond to the standard 0–5 star SLIPTA bands, and the rationale for these cut-offs is not clearly justified.

Recommendations:

• Explicitly state whether analyses were conducted using raw scores, percentages of the maximum score, or star equivalents.

• Clearly link observed score ranges to the official SLIPTA star levels.

• Consider presenting a table or figure showing the distribution of laboratories across SLIPTA star categories.

• If non-standard low/moderate/high categories are retained, provide a clear justification for the chosen thresholds and apply them consistently throughout the Results and Discussion.

2. Significance, Originality, and Contribution to the Literature

The manuscript documents low levels of compliance with laboratory quality standards in a large metropolitan area and highlights staffing and phlebotomy gaps. However, similar SLIPTA-based assessments have been conducted in Ghana and other African settings, and the added value of this study is not sufficiently articulated.

Recommendations:

• Clearly state in the Introduction what is new about this study (e.g., census coverage, detailed analysis of quality system essentials, explicit linkage to staffing norms).

• Avoid overstating uniqueness; instead, emphasize the contextual and operational contributions to Ghana’s laboratory network and policy implementation.

3. Study Setting, Sampling Frame, and External Validity

While a census approach is reported, the total number of eligible laboratories and the source of the sampling frame are not clearly described. There is also inconsistency between the stated inclusion of hospital laboratories and the reporting of private standalone laboratories in the Results.

Additionally, only one tertiary facility was included, while private facilities dominate the sample.

Recommendations:

• Clearly describe how the sampling frame was constructed (e.g., Ghana Health Service lists, regulatory registers, professional councils).

• State the total number of eligible laboratories by type and ownership and indicate whether any facilities were excluded or declined participation.

• Expand the limitations section to discuss how the facility mix and limited number of tertiary laboratories affect generalizability.

4. Measurement Tool (SLIPTA)

The introduction of non-standard compliance categories (low, moderate, high) does not align directly with SLIPTA’s official 0–5 star system, and the rationale for these categories is unclear.

Recommendations:

• Justify the use of these thresholds or consider reporting only continuous scores and standard SLIPTA star categories.

• Clearly state that the maximum possible SLIPTA score is 275 and relate observed scores to the official star levels.

5. Data Collection Procedures and Quality Assurance

The use of document review, observation, and structured questionnaires is appropriate. However, the manuscript lacks details on assessor training, standardization procedures, and quality assurance measures.

Recommendations:

• Report the number and qualifications of assessors.

• Describe any training or calibration exercises conducted.

• Indicate whether inter-rater reliability was assessed; if not, acknowledge this as a limitation.

• Clarify whether assessments were announced or unannounced and whether procedures were applied uniformly across facilities.

6. Human Resource Norms and Staffing Analysis

The comparison between expected and available laboratory personnel is informative, but the derivation of expected staffing levels is insufficiently explained.

Recommendations:

• Clearly state the source of staffing norms (e.g., Ministry of Health document and year).

• Describe how norms were applied across different facility types, including private and CHAG facilities.

• Clarify assumptions or adaptations and discuss how they may affect interpretation of staffing deficits and surpluses.

7. Statistical Analysis and Reporting

The use of non-parametric tests is appropriate; however, several statistical reporting issues need correction.

Recommendations:

• Report p-values appropriately (e.g., p < 0.001 instead of p = 0.00).

• Specify whether post-hoc pairwise comparisons were conducted following significant Kruskal–Wallis tests and report them if available.

• Where feasible, include measures of effect size or clearer interpretation of the magnitude of differences.

• Acknowledge the potential for type I error inflation due to multiple comparisons.

8. Interpretation Beyond Collected Data

The Discussion proposes explanations related to staff attitudes and resource allocation that were not directly measured in this study.

Recommendations:

• Clearly distinguish empirical findings from interpretations informed by the literature.

• Use cautious, non-causal language (e.g., “may reflect,” “could be attributable to”).

• Ensure appropriate citations accompany interpretive statements.

9. Ethics and Consistency of Reporting

There is inconsistency between the ethics information in the manuscript and the submission metadata, which indicates “N/A.”

Recommendation:

• Harmonize ethics reporting across the manuscript and submission system and briefly explain why ethical approval was required despite the absence of human participants or identifiable personal data.

Minor Comments

1. Abstract: Include key numerical results (median compliance score and range) and explicitly state that none of the laboratories met the minimum SLIPTA compliance threshold of 150 points, as this is a key finding.

2. Abbreviations: Define abbreviations at first use or include footnotes in tables (e.g., LMICs, UHC, SDGs, CHAG, NCCPA).

3. Terminology: Use consistent terminology throughout the manuscript (e.g., “quality laboratory standards,” “standard laboratory practice,” “quality management systems”).

4. Tables: Improve readability by standardizing formatting, indicating whether values are medians or means, and stating maximum possible scores for each domain.

5. Discussion: Correct the duplicated citation “(Mbah, 2014).”

6. Language and Formatting: Address minor grammatical and typographical issues and ensure references follow PLOS ONE style.

Strengths of the Manuscript

• Use of a validated international assessment framework (WHO SLIPTA).

• Census-based facility assessment, reducing selection bias.

• Detailed examination of quality system essentials and human resource distribution.

• High relevance for laboratory quality improvement and health policy in LMICs.

Reviewer #3: The manuscript addresses an important and policy-relevant public health issue i.e. compliance with quality laboratory standards in a resource-limited setting. The use of the WHO SLIPTA checklist, which is a globally recognized assessment tool aligned with ISO 15189, strengthens the methodological foundation of the study. The findings are clearly presented and consistent with evidence from similar low- and middle-income country (LMIC) contexts.

Overall, the study is within the scope of PLOS ONE and demonstrates scientific soundness. However, the following Major and Minor revisions are required to improve clarity, methodological transparency, internal consistency, and compliance with PLOS ONE reporting standards.

Major Revisions

1. Study Design and Sampling Clarification

• The manuscript states that a “census of all medical laboratories” was conducted, and 43 laboratories were assessed out of approximately 215 health facilities in the metropolis. But, it is unclear:

o How many laboratories actually exist in Kumasi

o Whether all laboratories were eligible

o Whether any laboratories declined participation

Hence I recommend, the authors should make the following Clarifications to enhance external validity and reproducibility:

o The total number of eligible laboratories should be stated

o Include the Inclusion and exclusion criteria

o Whether non-participation occurred and how it was handled

2. Inconsistency in Ethical Considerations.

There is a contradiction between sections:

• The Ethics section states that “this part of the study did not involve human participants”

• However, the study involved:

o Review of personnel files

o Human resource assessments

o Facility-level operational practices

These constitute institutional human data, even if not individual-level.

My Recommendations are:

• Clarify why ethical approval was required but human participation was considered “not involved”

• Explicitly state:

o Whether consent was obtained from facility management

o How confidentiality of institutional data was maintained

This is important for PLOS ONE’s ethics compliance.

3. Scoring Interpretation and Reporting

• The WHO SLIPTA checklist total score is 275, yet the maximum observed score is 104

• This creates confusion unless clearly contextualized

My Recommendations is:

Explicitly state:

• That all laboratories fell within the “No Star” category

• That compliance scores are presented as absolute values, not star levels

• Consider adding percentage compliance to improve interpretability

Example:

“The highest score (104/275) corresponds to 37.8% compliance.”

4. Statistical Reporting Issues

• P-values are reported as p = 0.00, which is statistically incorrect

My Recommendation is:

• Replace with p < 0.001, in line with reporting standards

5. Human Resource Analysis – Methodological Justification Needed

The “expected number” of personnel is described as estimated, but:

• The source of staffing norms is not clearly defined

• Some values (e.g., private laboratories expected to have zero scientists) appear unrealistic

My Recommendations are:

• Clearly cite the national staffing norms or policy documents used

• Explain assumptions used to estimate expected personnel numbers

• Clarify whether workload indicators (e.g., test volumes) were considered

Minor Revisions

1. Language and Style Improvements

• Minor grammatical and stylistic issues are present (e.g., spacing, punctuation, inconsistent capitalization)

• Quotations (e.g., Mahatma Gandhi) may be unnecessary for a scientific manuscript

I recommend that the authors should:

• Perform professional language editing

• Remove or reframe non-scientific quotations unless strongly justified

2. Tables and Presentation

• Some tables (especially Tables 5 and 6) are dense and difficult to interpret

• Abbreviations (e.g., NCCPA, CHAG) should be defined consistently in table footnotes

My recommendations are:

• Improve table formatting

• Consider summarizing key findings graphically (optional)

3. Discussion Section – Strong but Needs Tightening

The discussion is well-referenced and contextualized but could be improved by:

• Reducing repetition

• Clearly separating:

o Findings

o Interpretation

o Policy implications

I recommend the authors should:

Add a brief subsection on policy and system-level implications, particularly for:

• Ghana Health Service

• Laboratory regulatory authorities

• Accreditation scaling strategies

Reviewer #4: As a general comment there is vague words needs to acronyms such as, CHAG, NCCP, CLSI and else. Data avilability statement is not mentioned. Repetitive citation of Reference number 21 in one paragraph at line 35 & 36, similarly redundency of citaion at line 83 & 85. Operational definitions of some ambiguous terminologies of staff resource part such as, deficit, surplus and adequate

Reviewer #5: REVIEW COMMENTS

1. The manuscript should be reviewed for grammatical errors and coherence as few of them exist.

2. The authors should explain how the 43 hospitals and standalone laboratories were recruited to ensure probability sampling.

3. Discussion Line 19-20 attempts to justify private sector having poor infrastructure and obsolete equipment. However, context is important. Authors should indicate where those two referenced studies were conducted for better understanding.

4. Authors should consider using linking words (e.g. similarly, additionally, Also, on the other hand...etc) to connect statements in Line 34 to previous ones.

5. Line 50, the statement "Medical laboratory services contribute about 70%..." should be referenced.

6. Line 83-85 has reference Mbah (2014) appearing at both beginning and the end of the sentence. Authors should rectify that.

7. In conclusion (Line 96-98), "basic quality standards is concerning......" needs revision

6. PLOS authors have the option to publish the peer review history of their article (what does this mean? ). If published, this will include your full peer review and any attached files.

**Do you want your identity to be public for this peer review?** For information about this choice, including consent withdrawal, please see our Privacy Policy .

Reviewer #1: No

Reviewer #2: **Yes:** Rodas Getachew Abera

Reviewer #3: **Yes:** Lawan Adamu

Reviewer #4: No

Reviewer #5: **Yes:** Albert Abaka-Yawson

---

## [Author Response · Author response to Decision Letter 1]

21 Feb 2026

Response to Reviewers

Manuscript ID: PONE-D-25-53279

Title: Assessment of compliance with quality laboratory standards in medical laboratories within the Kumasi Metropolis, Ghana

Journal: PLOS ONE

We would like to express our sincere gratitude to the Editor and Reviewers for their thorough, constructive, and insightful feedback. We appreciate their recognition of the importance of our study in improving laboratory systems in low- and middle-income countries. The manuscript has been meticulously revised to address all major and minor comments. Below, we provide a detailed, point-by-point response to each issue, outlining how we have incorporated their suggestions into the revised manuscript.

Reviewer COMMENTS RESPONSES

Academic Editor

Abstract level revisions

1. Corresponding author The corresponding author has been indicated on the title page of the manuscript.

2. Facility Ownership and Classification The Abstract has been revised to explicitly state the number of public and private standalone laboratories assessed.

Location: Abstract (lines 26-28)

3.

Definition of Abbreviations All abbreviations (e.g., SLIPTA, CHAG) are now defined at first mention.

Location: Abstract (lines 26-29)

4. SLIPTA Scoring Method The Methods section now clearly explains how the scoring was done

Location: Methods (lines 30-34)

5. Reporting of p-values All previously reported p-values of p = 0.00 have been corrected and are now reported as p < 0.001, in line with statistical reporting standards throughout the manuscript.

6. Alignment of Findings and Conclusions The revised manuscript now explicitly states the expected score and its equivalent star rating; hence, the conclusion has been revised to align strictly with these findings.

Location: method (Lines 30-34); Conclusion (lines 42-43)

Introduction

1. Definition of abbreviations All abbreviations are now defined at first mention in the Introduction.

2. Reference 10 and substantiation of Claims Reference 10 has been corrected, and the statement is now supported by verifiable sources.

Location: Introduction (lines 92-34); References

3. Accreditation Institutions The Introduction now explicitly lists relevant accreditation bodies.

Location: Introduction (lines 91-92)

4. Study Gap and Problem Statement The Introduction has been strengthened to clearly articulate the knowledge and policy gap addressed by this study, specifically the lack of census-based data, including a detailed analysis of quality system essentials and staffing norms.

Location: Introduction (lines 85-96 and final paragraph)

Results

1. Expected MLS in Private Laboratories We have clarified that the expected staffing levels for private laboratories were derived from adapted national health sector norms, and this explanation is now stated in the Methods and the table footnotes.

Location: Methods: Data collection (lines 164-171); Table 3 footnote

2. Statistical support for facility-level performance A non-parametric pairwise comparison test has been performed to determine the best-performing facility.

Location: Result: Difference in laboratory compliance score by facility type (lines 241-255)

3. Column Clarity Explanatory footnotes, including raw scores and average scores as applicable, have been added to tables.

Location: Table 6(foot note)

4. Reporting of p-values All p-values in the Results section are now consistently reported as p < 0.001, where appropriate.

5. Discussion The discussion has been revised to align with updated statistical analyses, ensuring that interpretations reflect the revised results without unsupported inferences.

6. Reference 10 Spelling and Accuracy The bibliographic details for Reference 10 have been verified and corrected appropriately

Location: Introduction (Lines 85-87); Reference 19 and 20

Reviewers

1. SLIPTA Scoring, Categories, and Interpretation.

“The manuscript reports overall and domain-specific compliance scores (e.g., a median score of 61) without clearly explaining how these relate to the maximum SLIPTA score of 275 or to the official SLIPTA star-rating system”.

The relationship between observed scores and the WHO SLIPTA system has been clearly established. A new table presenting the distribution of laboratories across SLIPTA star categories has been added. Non-standardized score ratings have been expunged from the manuscript.

Location:

Methods: Instrument (lines 138-142); Data analysis (Lines 176-185).

Results: Laboratory compliance scores by facility type (ines 217-218)

Table 5: Laboratory compliance scores by facility type

2. Significance, Originality, and Contribution to the Literature.

The study's added value relative to existing SLIPTA-based assessments is not sufficiently articulated.

The Introduction has been strengthened to clearly articulate the knowledge and policy gap addressed by this study, specifically the lack of census-based data, including a detailed analysis of quality system essentials and staffing norms.

Location: Introduction (lines 85-96 and final paragraph)

3. Study Setting, Sampling Frame, and External Validity.

The sampling frame and facility composition are insufficiently described, and the implications for generalizability are not discussed We have clarified the sample frame and sampling procedures and included a discussion of their implications for external validity.

Location:

Method: Study population and sample (lines 124-128).

Strength and limitation (lines 335-342).

4. Measurement Tool (SLIPTA)

Non-standard compliance categories (low, moderate, high) do not align directly with SLIPTA’s official 0– star system The relationship between observed scores and the WHO SLIPTA system has been clearly established. Non-standardized score ratings have been expunged from the manuscript.

Location:

Methods: Instrument (lines 138-142); Data analysis (Lines 176-185).

Results: Laboratory compliance scores by facility type (lines 217-218)

Table 5: Laboratory compliance scores by facility type

5. Data Collection Procedures and Quality Assurance.

Details on assessor training, standardization, and quality assurance are lacking This has been clarified in the revised manuscript. The assessment was conducted by a licensed SLIPTA auditor.

Location:

Methods: Data collection (lines 150-152)

6.

Human Resource Norms and Staffing Analysis

The derivation and application of expected staffing levels are insufficiently explained

We have revised the staffing analysis for clarity and described how norms were applied across government, CHAG, and private facilities, including the assumptions underlying those applications.

Location: Data collection (Lines 164-169); Table 3(footnote).

7. Statistical Analysis and Reporting

Statistical reporting requires correction and clarification.

We have revised the statistical reporting throughout the manuscript. Post hoc pairwise comparisons following significant Kruskal–Wallis tests are explicitly reported. The potential for type I error inflation due to multiple comparisons is acknowledged.

Location

Methods: Data analysis (183-185).

Results: Table 8

Limitations (lines 335-342)

8. Interpretation Beyond Collected Data

The Discussion proposes explanations related to staff attitudes and resource allocation that were not directly measured in this study.

We have revised the discussion to clearly distinguish empirical findings from literature-informed interpretations.

9. Ethics and Consistency of Reporting

There is an inconsistency between the ethics information in the manuscript and the submission metadata, which indicates “N/A.”

We have harmonised ethics reporting; ethics approval details are now consistent across the manuscript and the submission system.

We are sincerely grateful to the Editor and Reviewers for the constructive feedback, which has greatly enhanced the clarity, rigour, and policy relevance of the manuscript. We hope the revised version now meets PLOS One’s publication criteria and look forward to your further consideration.

Sincerely,

Genevieve C. Aryeetey

---

## [Decision Letter · Decision Letter 1]

3 Mar 2026

Assessment of compliance with quality laboratory standards in medical laboratories within the Kumasi Metropolis, Ghana

PONE-D-25-53279R1

Dear Dr. Genevieve Cecilia Aryeetey,

We’re pleased to inform you that your manuscript has been judged scientifically suitable for publication and will be formally accepted for publication once it meets all outstanding technical requirements.

Kind regards,

Enoch Aninagyei, PhD

Academic Editor

PLOS One

Additional Editor Comments (optional):

Reviewers' comments:

Reviewer's Responses to Questions

**Comments to the Author**

1. If the authors have adequately addressed your comments raised in a previous round of review and you feel that this manuscript is now acceptable for publication, you may indicate that here to bypass the “Comments to the Author” section, enter your conflict of interest statement in the “Confidential to Editor” section, and submit your "Accept" recommendation.

Reviewer #2: All comments have been addressed

Reviewer #3: All comments have been addressed

Reviewer #5: All comments have been addressed

2. Is the manuscript technically sound, and do the data support the conclusions?

Reviewer #2: Yes

Reviewer #3: Yes

Reviewer #5: Yes

3. Has the statistical analysis been performed appropriately and rigorously? 

Reviewer #2: Yes

Reviewer #3: Yes

Reviewer #5: Yes

4. Have the authors made all data underlying the findings in their manuscript fully available?

Reviewer #2: Yes

Reviewer #3: Yes

Reviewer #5: No

5. Is the manuscript presented in an intelligible fashion and written in standard English?

Reviewer #2: Yes

Reviewer #3: Yes

Reviewer #5: Yes

6. Review Comments to the Author

Reviewer #2: The authors have satisfactorily addressed the review comments. The revised manuscript demonstrates improved clarity in the description and interpretation of SLIPTA scoring, appropriate alignment with the WHO star-rating framework, and strengthened reporting of statistical analyses. The Introduction and Discussion have been revised to better articulate the study’s contribution and to clearly distinguish empirical findings from interpretive commentary.

Reviewer #3: All substantive reviewer and editor concerns have been addressed.

The revised manuscript is methodologically rigorous, transparent, and ethically sound.

The study provides valuable, policy-relevant evidence without overstating its claims.

No additional analyses or clarifications are required.

The manuscript is ready for publication.

Reviewer #5: (No Response)

7. PLOS authors have the option to publish the peer review history of their article (what does this mean? ). If published, this will include your full peer review and any attached files.

**Do you want your identity to be public for this peer review?** For information about this choice, including consent withdrawal, please see our Privacy Policy .

Reviewer #2: **Yes:** Rodas Getachew Abera

Reviewer #3: **Yes:** Lawan Adamu

Reviewer #5: No
